# The Development of Telemedicine and eHealth in Surgery during the SARS-CoV-2 Pandemic

**DOI:** 10.3390/ijerph182211969

**Published:** 2021-11-15

**Authors:** Anas Taha, Bara Saad, Bassey Enodien, Marta Bachmann, Daniel M. Frey, Stephanie Taha-Mehlitz

**Affiliations:** 1Department of Surgery, Wetzikon Hospital, 8620 Wetzikon, Switzerland; bassey.enodien@gzo.ch (B.E.); marta.bachmann@gzo.ch (M.B.); daniel.frey@gzo.ch (D.M.F.); 2Department of Biomedical Engineering, Faculty of Medicine, University of Basel, 4123 Allschwil, Switzerland; 3Faculty of Medicine, St. George’s University of London, 2417 Nicosia, Cyprus; saad.b@live.sgul.ac.cy; 4Clarunis, Department of Visceral Surgery, University Centre for Gastrointestinal and Liver Diseases, St. Clara Hospital and University Hospital Basel, 4002 Basel, Switzerland; Stephanie.taha@clarunis.ch

**Keywords:** surgery, telemedicine, SARS-CoV-2, COVID-19, pandemic

## Abstract

SARS-CoV-2 has hampered healthcare systems worldwide, but some countries have found new opportunities and methods to combat it. In this study, we focused on the rapid growth of telemedicine during the pandemic around the world. We conducted a systematic literature review of all the articles published up to the present year, 2021, by following the requirements of the Preferred Reporting Items for Systematic Reviews and Meta-Analyses (PRISMA) framework. The data extracted comprised eHealth and telemedicine in surgery globally, and independently in Europe, the United States, and Switzerland. This review explicitly included fifty-nine studies. Out of all the articles included, none of them found that telemedicine causes poor outcomes in patients. Telemedicine has created a new path in the world of healthcare, revolutionizing how healthcare is delivered to patients and developing alternative methods for clinicians.

## 1. Introduction

The term telemedicine first appeared in the 1970s, describing treatment provided from a distance or far away using technology. The World Health Organization broadly defined telemedicine as providing health services from a distance by using information technology to obtain crucial data to spearhead diagnoses [1]. Additionally, the notion includes medication and treatment, prevention of disease, and research [1]. Physicians exclusively provided this service initially, but other health professionals began incorporating it into their practice; thus, the general term telehealth was coined [1]. All telemedicine is telehealth, but not all telehealth is telemedicine. Telemedicine in surgery reflects the use of telemedicine services when performing surgical procedures. In this manuscript, we used the term telemedicine in surgery to represent telesurgery.

Telemedicine has four basic principles: First, it provides virtual clinical support. Second, it is border- or geographical-barrier-free, meaning any person from any place in the world can use the service and receive access to healthcare facilities. Third, telemedicine has allowed hospitals to easily communicate to referral facilities for educational and consultation purposes, thereby reducing the overall cost of treatment to the patient and medical facility. Fourth, telemedicine has permitted the communication of medical centers and referral facilities; hence, enhanced efficiency care is delivered. The introduction and integration of telemedicine into health systems raised the standards of healthcare provided, especially in cases of emergencies. When the SARS-CoV-2 pandemic began in early 2020, it caught the world by surprise. Health sectors around the world were exhausted fighting the pandemic and were rapidly confronted by a lack of resources. Patients would avoid visiting the hospital due to fear of acquiring the virus, and healthcare providers limited the ability of patients to access the facilities they provide. Surgical services were also severely restrained during the peak of the pandemic. With preoperative triage and postoperative management becoming inherently riskier in person, healthcare providers and patients looked for other methods of communication.

At that juncture, our main goal was to explore the utilization of telemedicine and eHealth globally, and specifically in Europe, the United States, and Switzerland. We explored the benefits and limitations of telemedicine in many fields of medicine. We also examined how surgical services were hampered due to SARS-CoV-2, providing insight into the surgical activities most affected by the pandemic and how eHealth and telemedicine combated this. We first examined telemedicine in different disciplines before explicitly focusing on its use in various forms of surgery.

## 2. Methods

### 2.1. Search Strategy

This review began with conducting a literature search according to the requirements of the PRISMA approach through a free text search on the PubMed database on articles published up to 2021. The search terms adopted consisted of “telemedicine”, “telemedicine in surgery”, “eHealth”, “eHealth in surgery”, “COVID-19”, “surgery during the pandemic”, “teleconferencing”, and “teleconsultation”. We further integrated the Boolean operators to guarantee a vast search of the literature. Additionally, we conducted a manual search of the references used in the included studies and other reviews on the subject. Three researchers engaged in the search, and information extraction process individually. Minimal discrepancies arose (*n* = 2), but the whole team solved them through consensus.

### 2.2. Selection of Criteria and Evidence Quality

This study exclusively included original articles that had undergone peer-review. Moreover, these studies had to be in English and have defined telemedicine as in this manuscript. The authors also selected studies if they aimed to provide clinical guidance or educate surgeons and hospitals. The excluded studies were those that failed to address the topic. Nonetheless, the researchers avoided duplication analysis by choosing the latest publication from the same research group, saving time. The examination of evidence quality for this review occurred via assessing the reliability, usefulness, and authority.

### 2.3. Data Extraction

The authors conducted a full-text analysis of the included studies. The extraction of information involved the collaboration of three reviewers who assessed articles to determine if they addressed the research topic. The authors also evaluated the type of methods and the theories used by the research papers to gauge their reliability and relevance. After the data extraction process, the researchers stored the information on a laptop secured with a password. We solved any discrepancies (*n* = 1) by reviewing the articles again to establish a consensus. 

## 3. Results

The search on the PubMed database yielded 1536 results, while the free-hand search led to 10 studies. After the screening process, 101 articles remained, after which a further assessment identified 58 papers as appropriate for inclusion (Figure 1).

### 3.1. Multidisciplinary Use of Telemedicine

Telemedicine was helpful in previous outbreaks such as the SARS, MERS, Ebola, and Zika viruses [2,3]. Zhai et al. discussed how telemedicine allowed the assessment, diagnosis, management, and discharging of patients in isolation and hospital wards [4]. It also provided post-treatment care and follow-up consults. Before the pandemic, the American Medical Association reported that 74% of the American population did not have access to or were unaware of telemedicine [5]. However, in 2020, a survey of 842 physicians in the U.S. showed that nearly 50% of them used telemedicine during the pandemic compared to only 18% in 2018 [6]. In countries such as Switzerland, where the telemedicine ecosystem is relatively mature, in comparison to other European countries, the country’s largest private providers report about 2.5 million annual telemedical consultations, and that number is increasing [7,8]. Currently, several medical centers are performing triage via telemedicine, which allows for the screening of patients without the need for physical interaction [8]. Symptomatic patients use artificial-intelligence-driven chatbots, such as the Swiss–Italian project COVID-Guide, allowing them to rapidly self-assess their health status and receive recommendations on how to proceed [9]. Nittas and von Wyl discussed the general effectiveness and the cost-effectiveness of telemedicine. They stated that it has considerable potential for integration within the healthcare system during SARS-CoV-2 [8]. They also noted that the Swiss government sees the merit of telemedicine and is lobbying to receive more funding to expand the already well-established telemedicine system [8].

According to a study by Serper et al., who analyzed 1700 gastrointestinal/hepatology visits, 94% of visits occurred through telemedicine in the first four weeks of the pandemic [10]. A total of 88% of clinicians rated video visits as better or as good as face-to-face visits. According to the Commonwealth fund, 56% of behavioral health visits occur via telemedicine [11] (Figure 2). Uscher-Pines et al. reported that telemedicine allowed psychiatrists to gain insight into patients’ home settings and expanded their reach to previously underserved patients [12]. Most psychiatrists expressed that the transition to telemedicine was smooth and provided continuing care to patients who needed stability during the pandemic [12]. In pediatric psychiatry visits, Knopf described children as finding telepsychiatry easy to use and more comfortable [13]. Another large field using telemedicine during SARS-CoV-2 was endocrinology, with over 25% of their visits being performed virtually [11]. In Italy, the Society for Pediatric Endocrinology and Diabetology initiated telemedicine in one out of four health centers, using telemedicine for diabetic patient care during the pandemic [14]. Even in fields that require in-person assessments, telemedicine was a useful supplement. The Italian Society of Physical and Rehabilitation Medicine (SIMFER) pushed telemedicine to improve patient rehabilitation care during the pandemic [15]. SIMFER predicted that telemedicine will become the most effective method for rehabilitating chronically disabled people in the future [15]. In obstetrics and gynecology, a study of high-risk pregnancy patients showed that telemedicine for prenatal care is feasible and should be tailored for high-risk prenatal patients as it reduces SARS-CoV-2 exposure [16]. In oncology, studies have shown that telemedicine can provide quality care for patients, allowing them to be monitored remotely and reducing their risk of contracting SARS-CoV-2 [17,18]. 

### 3.2. Telemedicine in Surgery Globally

Ohannessian et al. claimed that Italy and France required several months before the respective ministries of health approved the reimbursements for telemedicine consultations [19]. A large systematic review analyzing 44 articles showed that telemedicine overall was easy to use, reduced patient waiting times, reduced travel expenses, improved communication with the doctor, reduced missed appointments, and increased access for those seeking medical care [20]. The implementation and integration of telemedicine into daily medical care are key factors in how countries will combat SARS-CoV-2 moving forward while maintaining care for those with other health needs [21]. Surgery was negatively affected during the pandemic. It was estimated that over 28 million surgeries were canceled or postponed worldwide between March and May 2020 [22]. From benign disease to cancer surgery, high proportions of surgeries were delayed. The global cumulative reduction in all surgery visits in 2020 was 16% [11]. It was estimated that Switzerland had a cancelation rate of over 9000 surgeries per week during the peak 12 weeks of the pandemic [22]. A hospital in Scotland reported a 58% decrease in admission into emergency medicine between March and May of 2020 in comparison to 2019 [23].

Bhaskar et al. posited that the massive urban–rural health gap witnessed in China makes it challenging to successfully implement the concept in surgery [24]. Puliyath et al. argued that India’s adoption of telemedicine in the surgery sector before the pandemic was minimal [25]. Bhatia claimed that telemedicine’s success in India is possible since healthcare facilities are primarily in urban areas [26]. In contrast, most Indian residents (up to 67% of the sample population) lived in rural destinations. In a study conducted by Thakurani and Gupta on 150 anesthetic surgeons in India, the COVID-19 pandemic forced these medical practitioners to transform and adapt their routines [27]. In Singapore, the incorporation of telemedicine in surgery during COVID-19 is likely to occur, as Bhaskar et al. reported that its use of the service has increased during the pandemic [24]. For example, in 2020, the region’s health minister stated that all chronically ill patients with Medisave qualified for teleconferences with their physicians, thereby minimizing the risk of exposure to the virus [24]. 

In Canada, El-Helou stated that telemedicine in spine surgery during the pandemic has increased [28]. Canada provides a great example of dealing with these issues. In the province of Ontario, a secure and publicly funded network called the Ontario Telemedicine Network (O.T.N.) is currently in use by hospitals and private practitioners. The network abides by the Canadian Personal Health Information Protection Act (PHIPA) by maintaining the patient’s privacy. In most cases, the Ontario Health Insurance Plan covers telemedicine on O.T.N. [29]. In the field of vascular surgery, the application of telemedicine is crucial to monitor clients that do not need to attend an in-person meeting [30]. Stoehr et al. found that the adoption of telemedicine in surgery is evident through the integration of the concept by worldwide surgical outreach initiatives [31]. The global adoption of telemedicine in pediatric surgery has received positive feedback. For instance, two surveys conducted by Metzger et al. showed that the adoption of telemedicine services before the pandemic was minimal [32]. Still, the prevalence of telemedicine during the outbreak has made people realize its advantages, such as minimizing stress for children. Bashshur et al. reported that the coronavirus pandemic has globally presented unprecedented chances that countries should not ignore [33].

### 3.3. Telemedicine in Surgery in Europe

The strongest motivation for the adoption and development of telemedicine during the pandemic has been the need to accommodate COVID-19 patients [24]. The congestion in various hospitals necessitated telemedicine services. Lakshin et al. revealed that only 21% of pediatric surgery branches in Germany provide telemedicine, but 51% of the departments began the service due to COVID-19, as illustrated in Figure 3 [34]. The study further revealed that 48% of the sample population stated that telemedicine was better than traditional visits. In contrast, 33% of the patients claimed that the service was worse than traditional visits. In the United Kingdom, the adoption of telemedicine services in the neurosurgery department has increased, particularly in tumor, functional, spine, and vascular cases. Of 10,746 patients included in the study by Mouchtouris et al., 7.1% of this population had integrated telemedicine visits into their routine [35]. Menendez et al. found that the United Kingdom has adopted a postoperative telemedicine platform that allows surgery patients to achieve their recovery journey with the support of practitioners through the PostopQRS application [36]. The app also allows surgeons to determine which patients are in dire need of face-to-face consultations.

Omboni posited that Italy has missed an opportunity to explore the benefits of telemedicine due to poor interconnection with patients [37]. The author also revealed that many government-funded telemedicine services are not cost-effective and seldom fit the adopted scientific evaluation models. In Northern Italy, the adoption of telemedicine services in plastic surgery has increased during the pandemic, and various doctors advocated for adopting the practice during COVID-19 [38]. Pignatti et al. further pointed that some conditions in plastic surgery are manageable via telemedicine [38]. In the Italian adult cardiovascular surgery sector, the surgeons had to stop elective surgeries to mitigate the spread of COVID-19. Donatelli et al. indicated a dire need for the healthcare system to be improved to ensure that people still have access to efficient healthcare after the pandemic [39]. Pinar et al. further argued that France eventually succeeded in integrating telemedicine into the urology division as 83% and 80% of the patients and physicians engaged in the study associated the concept with enhanced results, respectively [40]. In Denmark, replacing in-person meetings with telemedicine has rapidly increased, given the nature of COVID-19. Heeno et al. revealed that 85% of the sample population reported satisfaction with telephone consultations, although most preferred video sessions [41]. Another review by Pappot et al. showed that Denmark’s health practitioners provide training and maintain critical conditions, particularly those involving surgery patients [42].

### 3.4. Telemedicine in Surgery in the U.S.

Ghomrawi et al. reported that the United States division for Medicare and Medicaid developed laws requiring an equal number of in-person and telemedicine hospital visits [43]. The authors further argued that the development of telemedicine in surgical processes is likely to increase after the pandemic, even though these forms of visits often prevent a surgeon from correctly assessing surgical findings. Kichloo et al. showed that the United States acknowledged the importance of telemedicine during the pandemic, implying that the country will apply the concept even after COVID-19 due to its various advantages [44]. According to Bhaskar et al., the United States has increased patient access to telemedicine, where many insurers have opted to pay healthcare practitioners to deliver eHealth services to clients [24]. The government’s initiative of educating older people, who are skeptical of using technological devices, to receive services spearheaded the incorporation of virtual visits in various surgery departments in the U.S. [45]. Hurley et al. reported that eHealth services in orthopedic surgeries in the United States commenced in 1988 and have continued to progress to date [46]. Currently, surgeons use more advanced photograph technology that has consequently triggered improved accuracy and diagnoses. A review by Menendez et al. revealed a decreased adoption of formal physical therapy for people undergoing orthopedic surgery [36]. 

Dekker et al. revealed that the adoption of eHealth services in surgery in Colombia declined during the pandemic compared to before the coronavirus outbreak [47]. The fundamental causes for this trend include issues in cross-state licenses and lack of reimbursement, which eventually limited the successful deployment of telemedicine. In a review by Lanham et al., they found that 48% of American physicians began using telemedicine amid the pandemic [48]. Furthermore, 80% of the practitioners reported that they would dedicate efforts and investments to enhancing telemedicine during the coronavirus era. A cross-sectional survey by Salehi et al. revealed that in the field of facial and remodeling surgery, 91% of the surgeons that engaged in the survey acknowledged that they practiced telemedicine [49]. Of the participants, 75% stated that they would continue with telemedicine in the future, whereas 71% associated telemedicine with positive outcomes. Smith et al. illustrated that the United States eased some of the regulations on telemedicine due to the pandemic. The move has had positive impacts on the adoption of telemedicine in the U.S. [50]. For example, a study carried out by the Department of Veterans Affairs revealed that 95% of the people living in Greater Los Angeles found eHealth services to be very satisfactory [50]. Furthermore, De Biase et al. indicated that the prevalence of eHealth offerings in the neurosurgery department increased to 55% in April 2020 from 5% in the previous month [51].

### 3.5. Telemedicine in Surgery in Switzerland

In Switzerland, only 13% of the population was insured in an eHealth plan by one of the four largest private telemedicine providers [7]. Although these statistics may seem discouraging, they provide room for the country to increase its coverage and ensure all residents have access to telemedicine. A review conducted by Allaert et al. highlighted that Switzerland has struggled to develop telemedicine, as have all European states, but has demonstrated efforts to bridge the gap with the United States [52]. A retrospective analysis by Hübner et al. showed that many individuals from Western Switzerland had a positive attitude toward telemedicine as an excellent substitute for in-person visits [53]. The study’s shortcoming is that it failed to collect information on the particulars of the surgical consultations, such as the adoption of telemedicine [54]. Keshvardoost et al. stated that one of the biggest telemedicine centers is in Switzerland [55]. The institution is known as the Swiss Center for Telemedicine. The center’s existence means that the Swiss people will continue to receive these services after the COVID-19 pandemic. However, the use of telemedicine has been facing some challenges in Switzerland. For example, Seifert et al. found that the long-term care facilities in the country failed to provide Internet services for their residents, making it difficult to access telemedicine services [56].

A retrospective evaluation by Mamadnabiev et al. revealed that the acceptance of telemedicine in Switzerland is high due to the belief in Swiss surgical teams [57]. The study recommends developing telecommunication exchange networks through applications such as WhatsApp messenger to provide support. Such applications promote the availability of timely support and therapeutic reassurance, which enhances the quality of treatment. In other words, the future of telemedicine in Swiss surgical departments is promising, as the pandemic is teaching the country about the appropriate initiatives to integrate in the existing system. For instance, Contreras et al. claimed that providing 5G connections in rural areas will cause people living in urban areas to lag because the transmission of 5G signals does not travel extensive distances compared to 4G connections [58].

## 4. Discussion

The scale of the SARS-CoV-2 pandemic has triggered the rampant growth of telemedicine. With SARS-CoV-2 beginning in Wuhan in December 2019, the Chinese successfully implemented telemedicine to combat the disease [3]. Telemedicine became one of the safest methods for doctors and patients to interact by reducing physical proximity. Its benefits were not only limited to its safety. 

In many developing countries, the adoption of telemedicine, particularly in surgery, is not as advanced. For instance, Bhaskar et al. reported that in Africa, the use of telemedicine is in its infancy, and its incorporation into a post COVID-19 domain is uncertain [24]. During the pandemic, the number of patients opting for telemedicine services has increased since the first lockdown in the region in April 2021. These data imply that there are many opportunities for the introduction of telemedicine after the pandemic in India. For example, the authors noted the need for the reprisal and legalization of teleconsultation in anesthetic surgery [27]. The country has to adopt proper implementation directives to ensure the success of telemedicine.

Despite the program’s cancelation amidst the pandemic, different academic surgeons have incorporated alternative measures to reach out to patients. The primary approach integrated by surgeons is the use of telemedicine via various digital platforms. In addition, the opportunities require nations to conduct comprehensive studies for determining the best telemedicine and ones that are inappropriate. In the future, telemedicine ought to guarantee efficient and proper care after the crisis to safeguard the healthcare of future generations.

However, this new trend has some shortcomings, such as the possibility of information breaches through the adopted online channels. In the Caribbean, the application of telemedicine in surgery after the COVID-19 pandemic is uncertain due to the unavailability of comprehensive telemedicine systems and policies. For the successful integration of telemedicine in the Caribbean, the concerned parties should work on providing telemedicine solutions to the majority of the population, predominantly those residing in remote destinations [24]. 

In Europe, the adoption of telemedicine in pediatric surgery in Germany has already started receiving positive feedback, and this trend is likely to continue even after the pandemic [34]. The increased telemedicine adoption rate implies that the concept’s development after COVID-19 will reduce unnecessary in-persons visits, which was the preferred option before the pandemic. Additionally, the adoption of telemedicine in the surgical sector of Italy has faced challenges as a result of the overwhelming effects of the pandemic on the country [19,39]. These occurrences imply that the recognition of the need for telemedicine services has been high during the pandemic, and people are likely to accept the process in the post COVID-19 era. However, this trend will continue if the government provides the necessary resources to ensure the successful implementation of telemedicine.

In the U.S., the global COVID-19 pandemic has forced doctors to find alternative methods of attending to patients without necessarily meeting face-to-face to avoid the risk of exposure, which has significantly triggered the adoption of telemedicine and eHealth in the surgical field. In March 2020, the U.S. significantly expanded its eHealth system to accommodate more people through avenues such as FaceTime and Skype. It adopted virtual technology, the P.T. Genie, a wearable device that allows monitoring and managing patient recovery. However, successful enhancement of telemedicine in surgery during COVID-19 requires involved parties to refine the routine integration of the process into the surgical workflow. Nonetheless, there are still issues that need immediate addressing to ensure eHealth services can be used in the future. For instance, studies should evaluate the cost-effectiveness of the approach used to triage surgical patients from those not requiring surgery.

The European Commission has enabled Switzerland to deal with the various challenges of eHealth applications, promoting debate on technological advancement for eHealth. The involvement of Switzerland in trauma tele-education years before the pandemic illustrates that it aims to develop the service during the COVID-19 world. A review by Marttos et al. demonstrated that Switzerland hospitals have been involved in weekly surgical teleconferences and reported their results after two years [54]. The practice findings showed that it was essential to combine real-time data transmission and interactive decisions to increase diagnostic accuracy.

In addition, eHealth institutions in Switzerland are experiencing some shortcomings. These challenges include inadequate infrastructures such as free Internet and mobile phones to access eHealth applications for widespread eHealth solutions. Nittas and Von Wyl reported that Switzerland is the only country in Europe that offers established eHealth amenities [8]. However, in some of the sizeable eHealth service providers in the country, the services are mostly restricted. Nonetheless, with the SARS-CoV-2 pandemic, eHealth services will be functional to help limit infection rates and travel for medical attention. Therefore, the country will have to consider ensuring that everyone will have access to fast and reliable wireless connections regardless of their destination. Developing a system for monitoring telemedicine is also crucial to ensure that Switzerland does not encounter unprecedented and unanticipated challenges. Ultimately, the country will have to engage in comprehensive studies and acquire insights from knowledgeable individuals before fully developing its telemedicine practices.

## 5. Conclusions

The coronavirus pandemic has prompted the high acceptance and recognition of telemedicine. Amidst the pandemic, healthcare systems have struggled to ration their resources and reduce in-patient visits in surgery. Additionally, global medical and surgical facilities worked to assist both new and existing patients with outpatient care. With a tremendous increase in demand, telemedicine ensures that patients worldwide receive ongoing healthcare in all its forms and capabilities. The global utilization of telemedicine in surgery is not as advanced in developing countries as in developed states. In Europe, the congestion of people in various hospitals has necessitated telemedicine services in the region, especially in pediatric surgeries and neurosurgery in Germany and the United Kingdom, respectively. Similarly, the United States of America dedicated efforts toward integrating telemedicine during the pandemic, particularly orthopedic and facial and remodeling surgeries. Furthermore, Switzerland owns one of the biggest telemedicine centers globally, contributing to the high use of eHealth and telemedicine in surgery. Based on these arguments, it remains evident that the three countries analyzed can develop telemedicine and eHealth services during and after the pandemic. Nevertheless, the relevant states must engage in comprehensive research on the sector to guarantee the early realization and mitigation of challenges. 

## Figures and Tables

**Figure 1 ijerph-18-11969-f001:**
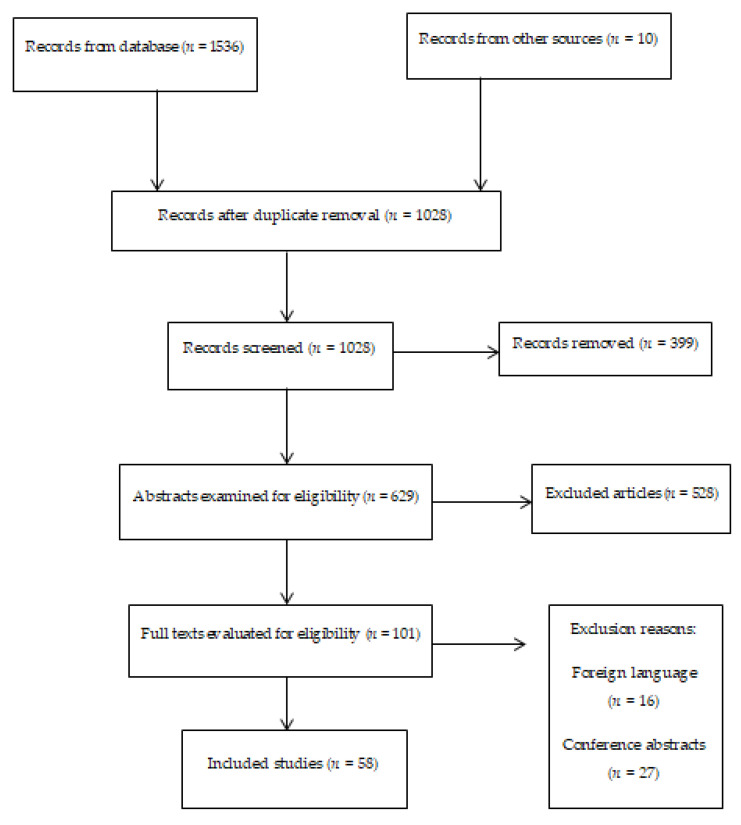
PRISMA flowchart.

**Figure 2 ijerph-18-11969-f002:**
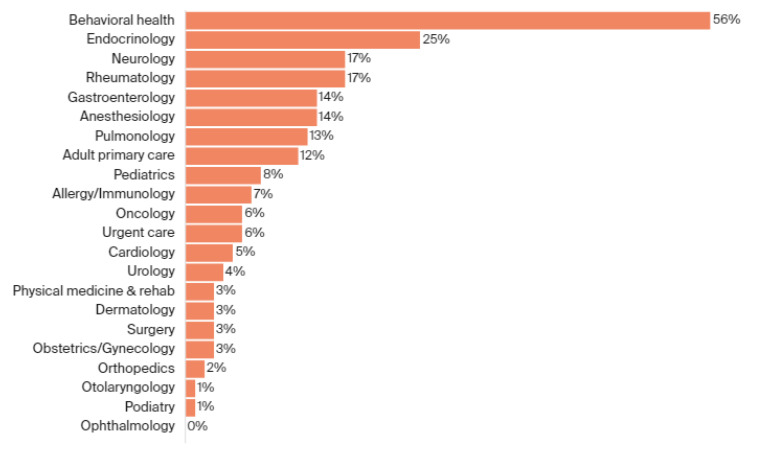
Telemedicine visits as a percentage of baseline. Data presented are expressed as a percentage. The number of telemedicine visits over the final three nonholiday weeks of 2020 is the numerator, while the number of visits in the baseline week 1–7 March multiplied by three is the denominator. Telemedicine includes both telephone and video visits. Shortened weeks or holidays were not included. (Reproduced with permission from Commonwealth Fund [11].)

**Figure 3 ijerph-18-11969-f003:**
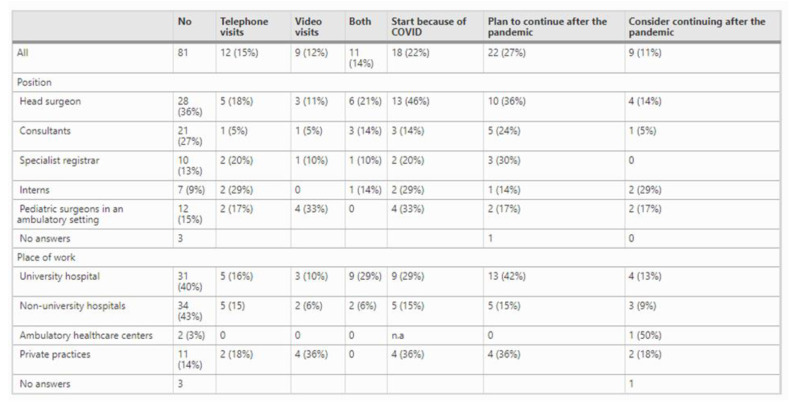
Telemedicine providence and professional features of research respondents (reprinted from Lakshin et al. [34]).

## Data Availability

The datasets used and/or analyzed during the current study are available from the corresponding author on reasonable request.

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
