# Peer review of "The Development of Telemedicine and eHealth in Surgery during the SARS-CoV-2 Pandemic"

_ijerph, 2021, doi:10.3390/ijerph182211969_

Round 1
Reviewer 1 Report
The manuscript titled ‘The Development of EHealth and Telemedicine in Surgery after the SARS-CoV-2 Pandemic’ addresses a highly relevant topic. E-health and telemedicine had been developing over the years. However, the irruption of SARS-CoV-2 with its impacts on postponing care of non-COVID-19 problems, on one hand, and reduced demand due to population’s fear to acquiring the virus, on the other hand, has strengthened the use of these new ways to provide health care services.
The manuscript, which is a review article, is both well organised and well written. The title of the manuscript describes a piece of work that addresses two areas: e-health and telemedicine in surgery. Although both of the areas are addressed in the paper, the main content is undoubtedly telemedicine in surgery (five out of seven sections are devoted to telemedicine in surgery). It is suggested then that the authors modify the paper’s title as to produce another that is more balanced to describe the contents of the manuscript. The abstract reflects the content of the article. There is general consistency among the different sections of the manuscript.
The authors do not explicitly state the aim of the manuscript. The aim of the paper is implicit in the abstract section (lines 11-13: ‘We explored the benefits and limitations of telemedicine in many fields of medicine. We also examined how surgical services were hampered due to SARS-CoV-2, providing insight into the surgical activities most affected by the pandemic and how Ehealth and telemedicine was used to combat it.’). In order to modulate the reader’s expectation on the paper, the aim of the manuscript should be explicitly stated at the end of the Introduction section.
The contents of the manuscript are very well developed. The organisation and depth of the contents are appropriate, progressing from a broader perspective on telemedicine in sections 1 (Introduction) and 2 (Multidisciplinary Use of Telemedicine around the World during COVID-19) to the main focus of the manuscript, which is telemedicine in surgery (sections 3 to 7).
Conclusions are provided in section 8 and they are consistent with the contents developed in the previous sections. However, conclusion have two weaknesses in the way they are currently presented. Firstly, conclusions refer to telemedicine (lines 322-323) to progress later to telemedicine in surgery (lines 323-325), which is correct because the authors are moving from the more general concept (telemedicine) to the more specific one (telemedicine in surgery) as they correctly do in the main text. However, the authors move back to the more general concept of telemedicine again (lines 326.329) which may be a little confusing. Secondly, in the last two phrases of the conclusion section the authors refer to the experience of Switzerland after referring to the adoption of telemedicine (the general concept) in the United States as compared to Europe. Instead of focusing on telemedicine in surgery (the more specific concept) to close the comparison between USA and Europe on the focus of the manuscript, the authors only refer to the experience of telemedicine in surgery in Switzerland. It is understandable that the experience of telemedicine in surgery in Switzerland be included in the conclusion section since it is one of the sections the authors develop in the main text (section 7). However, not mentioning the more general experience in Europe has the effect of leading to an unbalanced conclusion section as compared this section to the manuscript as a whole. It is suggested then that the authors make an effort to produce a conclusion section that corrects the described problems that affect the communicational effectiveness of a very good piece of research.
Author Response
We have edited your comments in the text, thanks for the feedback

Reviewer 2 Report
The authors present a personal description of the international use of telemedicine in surgery. The manuscript delivers an interesting picture of the situation worldwide. It is well readable and easy to follow. However, with regard to the best practice of research articles, the manuscript structure misses important elements. No information is given about material and methods in the main text. In the abstract, information about material, methods and results are missing, only the background and the conclusions are described. A discussion about the papers` topic is spread over chapters two to seven, accompanied by a brief introduction and very short conclusions. A main concern with this manuscript is therefore a potential bias, because the reader is not able to reproduce the intellectual process that lead to the findings. Furthermore, the reader could not distinguish between the conclusions of the cited literature - that would be results of a systematic review - and the interpretation of the authors - that would be the discussion in a systematic review. The authors seem to be very enthusiastic concerning the use of telemedicine in surgery. This raises concerns as well about the reliability and validity of the results. It is up to the journal to decide, whether they accept narratives as research articles.
However, there are major and minor weaknesses beyond this fundamental concern.
- Ideally, the authors would submit a systematic review that follows an internationally accepted structure. It is unclear, how the authors compiled the presented knowledge.
- The authors introduce a differentiation between telemedicine and telehealth. Later on, they us both terms. The authors should clarify, whether they use both terms later on synonymously or whether the terms always hold the definition introduced in the introduction. In case of the latter, the relevance of the statements concerning telehealth must be questioned in view of the paper’s title „telemedicine“.
- The authors speak about a situation „after“ the Corona pandemic. The reviewer thinks that the world is still in the middle of the pandemic. The incidences raise again even in developed countries. The author should clarify their definition of „after“ or alternatively skip this notion.
- A definition of „telemedicine in surgery“ is missing. The reviewer expected a paper about „telesurgery“ including teleassisted surgery. However, the examples are not specific for surgery. The authors should clarify, what they mean with „telemedicine in surgery“. Furthermore, they should explain why they excluded telesurgery. For example, the first paper listed in Medline defines telesurgery as providing „safe and accurate surgical procedures for patients who are unable to travel a long-distance.” (Choi et al., 2018).
- The reviewer disagrees with the statement on developing countries in line 32. In the contrary, telemedicine came up in developed countries like Australia or Scandinavia with a high level of health care on the one hand but wide empty spaces on the other hand. The authors state correctly in line 116, that the adoption of telemedicine in surgery is not as advanced in many developing countries.
- In the introduction, four principles of telemedicine are mentioned. But, the points three and four are not principles of telemedicine. A reduction of costs and an increase in the efficiency of health services as well as an improvement of the health care system are demands of the society towards telemedicine. If the reviewer understands the motivation of the authors correctly, it is the task of the paper to discuss the fulfillment of those demands by telemedicine in surgery. This part should be rewritten.
- The authors state, „As hospital beds around the world were filled to their maximum, capacity was reduced to almost zero for clinical patients“. Due to different answers and strategies of health policy towards the Corona pandemic, this statement is not true for all countries. Some countries reduced capacities for „normal“ patients to provide free beds for patients suffering from COVID-19. However, those beds were not always needed in those states. Moreover, the reduction of capacities might have corrected an oversupply of health services in some developed countries. The authors should be careful in repeating banal notions of the pandemic.
- Abbreviations should be written in full the first time (e.g. GI and LTCF).
- The statement in line 135 „For example, in 2020 ...” should be motivated by a literature reference.
- A link to figure 1 should be added to the main text.
- The superfluous space between „post-“ and „SARS“ in „post- SARS-CoV-2“ should be eliminated. Furthermore, the reviewer thinks that the world is not „post“ the pandemic.
- The reviewer could not follow the second sentence „The World Health Organization broadly defined telemedicine as providing health services from a distance, using information technology to obtain the information important for diagnoses, providing medication and treatment, prevention of disease, and research.“ Something seems to be wrong with the grammar. The reviewer recommends rewriting this sentence.
Author Response
Dear reviewer,
we have edited the manuscript according to your comments.
thanks for the feedback.
Sincerely,
Anas Taha

Round 2
Reviewer 2 Report
Thanks to the authors for considering the recommendations made in the first review. Only a few minor points remain.
- The reviewer did not find a reference to figure 2 in the main text.
- An arrow is missing between the two lower boxes in figure 1.
- Line 179: Two dots in series.
Author Response
Dear Reviewer
Thank you very mutch for your feedback. we processed your Comment. and I would like to inform you, that we have added Daniel M. Frey to the Authorsliste.
Anas Taha